# Exploring healthcare staff experiences with a hybrid paper/digital health management information system and their perspectives on digitalization as an alternative – A Tanzanian qualitative case study on perinatal data

Mary Cronin[1]*, Lucy Munishi[2], Gaudensia A. Olomi[3,4], Modesta Mitao[2], Blandina T. Mmbaga[2,3,5], Jackline Somi[2], Jairy Khanga[3,4], Ali S. Khashan[1,6], Francis M. Pima[2]☯, Simon Woodworth[6,7]☯*

1 School of Public Health, University College Cork, Cork, Ireland, 2 Kilimanjaro Clinical Research Institute, Kilimanjaro, Tanzania, 3 Kilimanjaro Christian Medical University College, Kilimanjaro, Tanzania, 4 Regional Administrative Secretary, Regional Health Management Team, Kilimanjaro Region, Moshi, Tanzania, 5 Kilimanjaro Christian Medical Centre, Kilimanjaro, Tanzania, 6 INFANT Research Centre, University College Cork, Cork, Ireland, 7 Cork University Business School, University College Cork, Cork, Ireland

☯ These authors contributed equally to this work.
* mary.cronin@ucc.ie (MC); s.woodworth@ucc.ie (SW)

## Abstract

Quality health data is essential to improve delivery and outcomes of healthcare. This study explores the experiences of healthcare staff in Kilimanjaro, Tanzania, using a hybrid paper and digital Health Management Information System, and their perspectives on transitioning to a fully digital system. It aims to understand current practices of perinatal data collection and utilisation and gather recommendations regarding the possible introduction of a fully digital HMIS (DHMIS). A case study design was employed; individual semi-structured interviews were undertaken with staff from four professions directly involved in data generation and use ($n = 29$), working in a range of healthcare settings. Thematic analysis was conducted using NVivo 12 software; it identified findings under four major themes, along with a series of recommendations on the implementation of the DHMIS. We found that while in theory the hybrid paper and digital system facilitates standardised data management, in practice it presents inefficiencies in manual data entry leading to challenges with data accuracy, loss, retrieval, storage and flow. These challenges contribute to the strongly positive attitude among healthcare staff towards adopting a DHMIS which they believe would improve data accuracy, reduce workload, and enhance clinical and policy decision-making. To achieve a successful DHMIS, participants recommended effective training for all users. Additionally, they proposed an integrated system to avoid data redundancy. The importance of robust infrastructure to ensure sustainability, and of reliable internet and electricity supply, were also highlighted. In conclusion, this

**Data availability statement:** All relevant data for this study are publicly available from the Zenodo repository (https://doi.org/10.5281/zenodo.14287006).

**Funding:** The ULTRA project received funding from the following: The Global Pregnancy Collaboration (CoLab) funding to develop prototype ULTRA app - ASK. https://pregnancycolab.tghn.org/ Irish Research Council Coalesce Award, funded by the Irish Department of Foreign Affairs (Irish Research Council: COALESCE/2021/51). PI - ASK and Co-PI - BTM. https://research.ie/ The funders had no role in study design, data collection and analysis, decision to publish, or preparation of the manuscript.

**Competing interests:** The authors have declared that no competing interests exist.

study provides valuable insights into the shortcomings of paper-based perinatal data recording, and the potential benefits and challenges of implementing a DHMIS in low-resource settings. It underscores the necessity of strategic planning, investment in infrastructure, and capacity building to achieve successful digital transformation in healthcare. The findings align with global health strategies promoting digitalisation to enhance health outcomes and support data-driven decision-making in healthcare systems.

## Introduction

Quality health data is essential to monitor, evaluate, prioritise, and improve the delivery of healthcare services [1]. In 2018, the World Health Organisation (WHO) 71st World Health Assembly passed a resolution on digital health exhorting member states to prioritise adoption and implementation of digital health technologies to support achievement of national targets, the Sustainable Development Goals and Universal Health Care [2]. Furthermore, for Low- and Middle-Income Countries it is considered important to develop an organisational culture which emphasises data-informed decision-making [3]. The WHO Global Strategy on Digital Health 2020–2025 [4] emphasises health data should be categorized as sensitive personal data; this calls for extremely high standards of security and necessitates cybersecurity, building trust, accountability and governance, ethics, equity, capacity building, and literacy, as well as ensuring high-quality data are gathered [5].

There is considerable evidence that digital health interventions are feasible and sustainable in low-resource settings [6,7]. At the same time many factors are known to impede or promote investment, sustainability, and scale up of digital technology interventions. Reviews on Low- and Middle-Income Countries have identified facilitating factors such as strong stakeholders' commitment and involvement, government support, intersectoral and interinstitutional collaboration, networking and collaboration with other implementing partners, improved client satisfaction, experience, and confidence in using the system, motivation, and competence of staff [8]. Specific barriers included limited (poor) infrastructures, low internet connectivity, and unreliable electricity [8] along with psychological and personal barriers and increased workload among health care workers [9]. Hesitancy to transition fully to digital systems has been reported among healthcare staff mainly due to perceived risks about the sustainability and funding of electronic systems [10]. Proposed solutions emphasise training and providing educational programs to enable staff fully utilise the technology [9].

Karamagi *et al.'s* comprehensive review on the use of digital health in the past ten years in sub-Saharan Africa highlighted how, despite large numbers (738) of functioning digital health interventions, they are poorly coordinated, not integrated into the countries' systems, investments in digital health not equally distributed, and lack planning for sustainability and scaling up of the feasible interventions [11]. They argue that *"Successful implementation of digital health interventions requires a balance*

*of a good digital tool, a receptive user, and an ideal context at the policy and implementation level"* [11]. Mugauri *et al.'s* [12] recent scoping review of the implementation of electronic health records in eight sub-Sahara African countries from 2014–2024, reported benefits such as improved data quality and accessibility, contributing to better-informed patient care. Challenges noted included infrastructure deficits, financial limitations, and data privacy issues, while difficulties with staff acceptance of new systems were reported as a critical barrier.

Many of these challenges are echoed in recent Tanzanian research. For example, Mwogosi *et al.'s* [13] narrative review on the integration of internet and artificial intelligence technologies in primary care, highlighted the necessity of capacity building and resilient digital infrastructure, along with the merits of systems design involving end users. Mwogosi and Kibusi [14], in their qualitative study with 14 participants (administrators and healthcare professionals), exploring barriers to the implementation of electronic health records (EHRs), reported operational challenges arising from poor management support and a greater workload. Additionally, inadequate digital infrastructure, a shortage of trained technical personnel and system maintenance requirements were considered a risk to health service quality and patient outcomes.

Tanzania has a highly standardised national health management information system which has evolved in recent decades. In 2013 it introduced a hybrid health management information system (HMIS) combining updated paper-based records with digitalisation in the form of DHIS2, which had been customised for compatibility with the paper-based system [15,16]; the goal was to promote ease of data access and utilisation [17]. For the past ten years this has led to digital data management at district level and higher, while lower-level facilities, such as dispensaries, have continued to use printed paper-based data registers and forms [3,16]. Within perinatal care data standardisation is achieved through the distribution and use of a common set of printed ledgers, known as MTUHA books, in which antenatal, labour and delivery, postnatal and early child development data are recorded [3,18]; variation from these government mandated books is not accommodated.

This level of standardisation should mean MTUHA books data provide robust primary data which can be aggregated and submitted to inform district, regional and national level decision-making, however there is evidence to the contrary. System use has been described as weak, with routine data at district level found to be inaccurate [1] and of mixed quality [19]. It has been suggested that shortcomings in data quality derive, in part, from errors when manually summarising and copying data into paper-based reports [15,20]. Additionally, it has been reported that hospital management in maternity wards has introduced supplementary registers, along with nurses creating informal, often narrative, documentation for various unofficial purposes, resulting in a HMIS which is complex and prompts, or requires, duplication of data [3,21]. Also, nurses were reported as alienated from quantitative HMIS data systems, describing themselves as "mere data producers", and not finding this type of data useful in their daily clinical work [3, p. 6]. Accurate data collection and reporting is also negatively impacted by challenging work environments including chronic shortages of healthcare staff, especially in rural areas where approximately 70% of the population lives [22], while repeated reallocation of staff can lead to loss of institutional knowledge [3]. Both factors can contribute to staff finding it challenging to record data at the time of collection and instead recording it at the end of a shift with the attendant risk of inaccuracy (ibid).

In contrast, where the digital DHIS2 system is in operation, district health managers have reported very high levels of workload reduction (81.72%) and enhanced data use (86.18%) [15]. However, there are still reported challenges including lack of, or slow, internet connections, inadequate information communications technology infrastructure, pressure within the system close to data submission deadlines, and inadequate technical support and system maintenance [15,16,23]. These problems all make it more difficult to transfer knowledge and build capacity [21].

**A case study of current and prospective digital perinatal HMIS**

This research aimed to better understand how perinatal data is collected, recorded, shared and used within the current hybrid HMIS in the Kilimanjaro region of Northern Tanzania, through exploring the perspectives of four professional staff groups directly involved – Nurse-Midwives, Obstetrician/Gynaecologists, District Medical Officers, and District

Reproductive and Child Health Co-ordinators – and gathering their recommendations on how to successfully introduce a fully-digitised perinatal data system.

Kilimanjaro is a largely rural region with seven districts namely Moshi District (rural), Moshi Municipal (urban), Siha, Mwanga, Same, Rombo, and Hai; it has a population of 1,861,934 and approximately 44,000 births annually [24]. A desk review at the regional health office indicated there are 427 healthcare facilities of which over 80% are dispensaries; all dispensaries provide ante-natal care, but only designated dispensaries with well-improved infrastructure are allowed to provide delivery services.

This case study was part of a multi-method project during which a digital maternal and child health registry app, ULTRA, developed within the DHIS2 platform, was piloted and evaluated in this region [25]. The ULTRA partnership involves several research centres, clinical institutes and health authorities in Kilimanjaro, Tanzania, University College Cork (UCC), Ireland and the Global Pregnancy Collaboration (CoLab) (https://pregnancycolab.tghn.org). This qualitative study involves formative research to explore the operation of the current hybrid system to inform the development of the ULTRA project and how best to successfully introduce and operationalise an effective, comprehensive digital system in the region. It is well aligned with Kumar and Mostafa's call in their landscape study on the role of electronic records for better health in LMICs, for in-depth research to offer guidance on the development of systems, health information architecture, organisational resources, interoperability and data standards, data quality and data use [2].

## Materials and methods

### Study design, sample and participants

An interpretive methodology was adopted for this case study [26, 27], with qualitative, individual, semi-structured in-person interviews [28] selected as the data collection method. The core team of four qualitative researchers (MC, SW, FP, and LM) collaborated closely on study design and implementation holding weekly online meetings for the duration of the project. Context-specific guidance was provided by co-authors MM, GAO, BTM and ASK from the wider ULTRA team.

Our sample was constructed to include professional staff groups with a role in data collection, recording, analysis, and utilisation in the seven districts in the Kilimanjaro region, involving both urban and rural contexts, and across a range of facilities (public, faith-owned and private). A purposive sampling strategy [29] was used. One Nurse-Midwife (NM), District Reproductive Child Health Co-ordinator (DRCHCo), and District Medical Officer (DMO) was recruited from each district, while an obstetrician/gynaecologist (OBGYN) was recruited from each district hospital of which there are 8, resulting in 29 experienced participants; (see Table 1 – Participant Information). Recruitment was supported by the Regional Nursing Officer (GAO) who oversees all health research activities in the region. Data saturation was achieved with this sample size.

### Reflexivity

The core team consisted of early career researchers in Kilimanjaro (FMP and LM), and experienced researchers in Ireland (SW and MC). FMP has a Master of Public Health degree and LM has a Bachelor of Arts in Community Economic Development; both are Tanzanian staff at KCRI. SW and MC are academics in UCC. SW has a B.Sc. in Computer Science with an M.Sc. and a Ph.D. in Management Information Systems; he has previous research experience in Malawi as well as an established track record in Connected Health and Health Informatics. MC is a social scientist and experienced qualitative researcher with qualifications in General and Sick Children's nursing; she has worked in two countries in sub-Saharan Africa. The core team reflected individually and collectively on their roles and positionality throughout and worked in a complementary manner to maximise the contribution of each member.

### Ethical considerations

Ethical clearance was secured from the Tanzanian National Institute for Medical Research (Log number NI MR/HQ/R.8a/Vo1. IX/4075), which also approved publication of this article. Data management complied with Irish data protection

**Table 1 . Participant Information.**

| ID Number | Date of Interview | Participant group | Duration |
|---|---|---|---|
| NM 01 | 21/11/2022 | Nurse/ Midwife (NM) | 36 min |
| NM 02 | 22/11/2022 | | 38 min |
| NM 03 | 01/12/2022 | | 37 min |
| NM 04 | 08/12/2022 | | 37 min |
| NM 05 | 14/12/2022 | | 39 min |
| NM 06 | 15/12/2022 | | 39 min |
| NM 07 | 15/12/2022 | | 49 min |
| OBGYN 01 | 14/12/2022 | Obstetrician/ Gynaecologist (OBGYN) | 29 min |
| OBGYN 02 | 14/12/2022 | | 16 min |
| OBGYN 03 | 05/01/2023 | | 26 min |
| OBGYN 04 | 31/01/2023 | | 44 min |
| OBGYN 05 | 10/02/2023 | | 30 min |
| OBGYN 06 | 10/02/2023 | | 31 min |
| OBGYN 07 | 02/03/2023 | | 30 min |
| OBGYN 08 | 10/03/2023 | | 26 min |
| DRCH 01 | 08/12/2022 | District Reproductive and Child Health Co-ordinator (DRCHCo) | 44 min |
| DRCH 02 | 06/01/2023 | | 34 min |
| DRCH 03 | 13/01/2023 | | 30 min |
| DRCH 04 | 20/01/2023 | | 35 min |
| DRCH 05 | 07/02/2023 | | 44 min |
| DRCH 06 | 02/02/2023 | | 36 min |
| DRCH 07 | 10/03/2023 | | 42 min |
| DMO 01 | 05/01/2023 | District Medical Officer (DMO) | 28 min |
| DMO 02 | 13/01/2023 | | 21 min |
| DMO 03 | 23/01/2023 | | 30 min |
| DMO 04 | 24/01/2023 | | 15 min |
| DMO 05 | 07/02/2023 | | 28 min |
| DMO 06 | 02/03/2023 | | 38 min |
| DMO 07 | 10/03/2023 | | 23 min |

**NOTE: Participants were drawn from 10 facilities across seven districts.**

legislation and Data Protection Guidelines in Tanzania. Prospective participants, none of whom had prior experience of the ULTRA app, received an information sheet in advance and had the opportunity to have questions answered. Those who volunteered to participate provided written consent which was witnessed by the interviewer. A secure, online, group account was created within UCCs data storage system and data was accessible only to the core qualitative study team. Participant quotes are anonymised, with a code indicating their professional group. This article is written in accordance with COREQ guidelines for reporting qualitative studies.

## Data collection

A topic guide was piloted with four staff, prompting minor amendments. Following consultations with participants, interviews were undertaken by appointment between 1st November 2022 and 30th March 2023 after work at participants' workplaces, in a location of their choice. Conducted in Kiswahili by FMP, with LM as notetaker, interviews were audio recorded and ranged in duration from 15 to 45 minutes. Participants were provided with a small financial remuneration of 500 Tanzanian Shillings in acknowledgement of their contribution.

### Data analysis

Interviews were transcribed verbatim in Kiswahili, translated into English, and uploaded to NVivo 12 software. Thematic analysis was undertaken using a codebook approach [30] and following Braun and Clarke's 6-step process [31]. Four transcripts were read and coded independently, and then collaboratively by two researchers (FMP and LM), following which a preliminary codebook was developed. These transcripts were also read by MC and SW and following discussion the codebook was further developed. During weekly meetings new codes were discussed as further transcripts were analysed, leading to refinement of the codebook. Analysis was completed using both a deductive approach based on pre-determined themes, and an inductive approach for generation of recommendations. The themes developed were (1) Perinatal Data Collection, Recording and Storage, and Challenges Arising; (2) Data Flow; (3) Data Usage; and (4) Perceptions and Expectations of a fully digitalised HMIS. The fifth theme presents staff recommendations for achieving a fully digitised system.

## Findings

### Data collection, recording and storage, and challenges arising

Data on an individual pregnant woman is collected by the Nurse Midwives and recorded in the 'Mama card' which the woman keeps and brings to her appointments. It is also recorded using one of the three approaches to facility-level data recording, with the approach in use depending on the level of health facility. The first approach, which is exclusively paper based using MTUHA books, operates in all government dispensaries and in some health centres also. For each MTUHA book data must be manually entered in three different books (registered book, tally sheets and report books). Sometimes, when MTUHA books are out of stock Nurse Midwives (NM) record data in 'counter books' and must later transfer them into the MTUHA books when again available. A second approach involves dual recording in both MTUHA books and the digital DHIS2 system: this operates at district, regional and national hospital levels, as well as in some designated health centres. A third approach, digital-only data recording, is used only in some private hospitals, where a variety of digital systems are employed.

The types of data collected and recorded are highly standardised across facilities. Interview data from NMs revealed a clear understanding of the clinical purposes of mandatory data recording, however, full and complete recording in the four MTUHA books was found to be time-consuming, especially when working alone as described below.

*"There are a lot of challenges, the writing becomes too much, you write in different books, there are many MITUHA books, so the pregnant mother is included in various books, so the process of including the mother in all the books is getting long, it is taking time, another challenge is tearing of the books."* (NM 01)

*"Now the challenge is that the service provider has many customers, and he is alone, he may forget to write in the register and write only in the pairing, so it is a challenge that you find that other parts are not filled."* (DRCHCo 01)

The use of 'counter books', and the necessity of transferring data later to MTUHA books when again available was identified by one District Medical Officer as creating extra work for the NMs.

It was reported that MTUHA books are to be stored by NMs in locked cabinets, typically in consulting rooms which are to be kept locked and secured when not in use; access is limited to NMs, medical record department administrators, and the medical officer in charge of the facility. However, challenges including data loss and difficulties with data retrieval were reported, arising from the use and storage of paper-based records, for example,

*"… the environment may not be friendly (as you may find different books stored in one box) to the extent that the books are damaged which is not safe, for storage becomes a bit of a challenge."* (DMO 07)

*"…. past information get lost easily and you see these put in the box, you come tomorrow to find that it has been eaten by ants or has been flooded."* (DRCHCo 01)

Concerns were raised by some participants who believed storage to be insecure as the person in charge might forget to close cabinets. Knowledge of correct storage of paper-based data differed between staff groups for example:

*"At the medical records there is a MTUHA focal person who is responsible for preserving the paper-based registries to the store; according to the guidelines they are to be preserved for five years."* (DRCHCo 03)

*"…The duration of storage in the cabinets or in the system is still unknown."* (NM 01)

### Data flow

A monthly report of aggregated data of perinatal health indicators is generated by each health facility. At dispensary level, the paper-based monthly report is manually prepared by a NM based on MTUHA books data. Where the dual approach is in place, summary report data is entered directly into DHIS2 by an NM. Full and accurate completion of paper-based monthly reports appears to be a challenge; a District Reproductive and Child Health Co-ordinator described how some are not properly filled or not filled at all. In addition, data sharing through these reports was described as 'expensive' and 'time consuming' often involving hand-delivery by a NM to the district office. As data on each individual pregnant woman is only recorded in the Mama card, and sometimes in a referral letter, it is not accessible at district level where only aggregate data is available.

The District Reproductive and Child Health Co-ordinator (DRCHCo) receives all aggregated data reports in their district, verifies their accuracy and completeness, and checks for missing data. When satisfied, the DRCHCo shares and reviews it with the District Medical Officer at which point the data is available to inform decision making at district level. Once in the DHIS2 system, data from each health facility is accessible to the Regional Medical Officer's office and Ministry of Health.

### Data usage

NMs report that the primary use of data is in the care of the individual pregnant woman, to support her wellbeing and appropriate management. Data is also used by obstetrician/gynaecologists to manage typical, and complex pregnancies, labour, and deliveries. Both NMs and obstetrician/gynaecologists (OBGYN) were also found to make practice-informed and data-informed recommendations to the District Medical Officer (DMO) during departmental and health management team meetings.

Monthly summary reports from individual health facilities are considered collectively at district level by the DMO, DRCHCos, and district level heads of clinical departments, who decide on and plan for interventions in areas that require attention.

*"For example, we have analysed the data and found that many mothers are late starting their first clinic visit under 12 weeks. Once you know they are late, you must start looking for the reason why they are late. Once you know, it is like you are going to wake up the service providers to find out what the challenge is and plan for immediate intervention."* (DRCHCo 04)

*"The information helps us to decide how to provide care for women with complications….."* (DRCHCo 02)

At district level, DMOs, who have the authority to approve, or otherwise, recommendations proposed by NMs, OBGYNs, and DRCHCOs, use data to inform clinical management, identifying gaps, needs or challenges (for example vaccination coverage, availability of drugs), and plan appropriate interventions.

*"We use the information collected to address challenges, by looking at the indicators we identify areas that need to be improved."* (DMO 04)

DMOs reported making data-informed recommendations for changes in policies and guidelines to the Regional Medical Officer, to the Ministry of Health, and to the ministry known as the 'President's Office, Regional Administration, and Local Government', in accordance with recommendations in the Tanzania Health Sector Strategic Plan July 2021-June 2026.

*"We can influence policy reforms into two categories. First, we can present our insights at the regional level through the RMO, since we are the persons helping him at the district level. Second, we normally have several meetings where we are invited by the Ministry of Health and through these gatherings, we air out our views and opinions."* (DMO 04)

Final decisions regarding policy or guideline amendments or implementation were described as a function of the ministries.

In summary, health professionals are very aware of ongoing challenges related to incomplete data recording, data loss, retrieval, and insecurity. Data quality is inconsistent but nonetheless it is used within clinical practice, to identify health service delivery issues, as well as informing resource allocation. Based on current experiences with data management, staff groups were found to have clear ideas regarding a full DHMIS.

## Perceptions and expectations of a fully digitalised HMIS

The prospect of a fully digital perinatal data system being introduced across all government-run health facilities in the Kilimanjaro region was regarded very positively across all four staff groups, while some concerns were also articulated. For OBGYNs and NMs as clinicians, the speedier, easier access to a patient's medical history offered by digital data was regarded positively as it is expected to facilitate better informed and, therefore, safer clinical care for women and their babies.

*"……the digital systems it has simplified our workload at a bigger percent, compared to the paper-based record system where there are many files hence you can waste the whole day looking for a patient's file. On the electronic system it is easy to retrieve patient's data it can only take a minute."* (OBGYN 05)

*"……Digital system is effective to use, you avoid the workload of filling in data into many MTUHA books compared to online where the process is quick, and it saves time."* (NM 05)

For DRCHCos, who currently input and manage much of the data it is perceived as a 'game changer' leading to faster recognition and addressing of challenges arising in practice settings.

*"This will enable health workers to input information data directly and thus data analysis becomes easier to follow through every week to realize the problems early and create immediate interventions….."* (DRCHCo 05)

*"…… I believe the project will be the biggest success in health information systems….."; "…. I hope it cuts across in all regions countrywide…."* (DRCHCo 04)

DMOs reported that in health facilities with DHIS2 already in operation the quality of services has improved efficiency, benefiting both clinical and policy decision-making, and providing daily access to district level data as well as individual patient data. One stated

*"So, in my experience digital registries are very efficient because you get daily reports and you are able to make prompt decisions, we don't wait for monthly decisions to be made."* (DMO 07)

No barriers to the introduction of a digital system were reported. While one participant suggested that young staff are already capable and ready to operate one, age was not identified as a barrier to staff being trained and capable of successful system use. Some notes of caution were also struck. For example, some participants had experienced previous attempts at the introduction of DHMIS and so they desired to see investment in a comprehensive, sustainable system to achieve long term, positive impact.

> *"……. Yes, I believe through digitisation services will be improved well but with caution ……… the government should invest in forming strong and sustainable systems which will improve efficiency and effectiveness of health services in the long run."* (DMO 07)

Staff currently working with digital systems raised concerns regarding the availability of support staff to efficiently manage technical issues, along with worries of system failure leading to the loss of valuable data, for example,

> *"…....Digital systems have many challenges, this include the malfunctioning of the systems, where you have to call the IT technician for repair and many of them are not easy to access on time, so activities are halted for several days."* (NM 05)

> *"……they fear the reliability of digital systems because sometimes the systems fail to function, so that is also another challenge to consider."* (DRCHCo 06)

Inconsistent internet access and electricity supplies were identified as concerns

> *"…… Another major challenge is the electricity cut offs and low network connection; this interrupts our daily work at the hospital…..."* (OBGYN 02)

## Participant recommendations for a fully digitalised HMIS

### Effective training

Effective training was identified as essential by all participants; it was recommended to be as widespread as possible across all staff groups with responsibility for data recording and analysis, to ensure successful adoption of the new system.

> *"…. I would suggest the systems should be present in every department and health workers should be well trained and competent to use the systems and everyone should be given access in case a person misses because personal emergencies do happen, then another worker can take over the activities".* (NM 05)

### Preference for an integrated system

DRCHCOs and DMOs wish for the introduction of an integrated system as, in some health facilities currently, and across the region, there are several non-integrated digital systems. Consequently, staff must enter the same data into different systems resulting in time-consuming data entry and slower data retrieval.

> *"…....I would suggest having a system that is integrated with all kinds of information concerning maternal and child care this would be ideal for the kind of workload health workers experience but it will also avoid repetitions from the MTUHA books to the digital registry, so when a nurse is aware that the data has been inputted there is no need for her to repeat the same information elsewhere."* (DRCHCo 05)

## Investing in system sustainability

In recognition of the reality of poor or inconsistent internet access, some participants recommended that any new system should support offline data recording with the functionality to upload data centrally once a day, as and when a mobile or WiFi network becomes available. The importance of system stability was also identified; if this isn't the case it was anticipated that staff could quickly seek to revert to the MTUHA books system.

*"In cases like these we normally go back to the paper-based record system to avoid data loss."* (DMO 02)

*"……Furthermore, the people responsible for making these systems should form stable and functional applications to avoid system failure because when a health worker encounters a malfunction of the system, they automatically retract from using digital systems to the paper-based record systems."* (DRCHCo 05)

The necessity and importance of thorough system preparation and set-up was explicitly identified; this is particularly important in facilities known to have inconsistent internet access or electricity supply.

*"……so, the people making these systems should thoroughly check, verify and pre-test these systems before they are dispensed to the end user."* (DMO 02)

One DMO captured the complexity of the prospective introduction of one, integrated, digital system,

*"…...but the systems need a lot of investment, because in scenarios where a system failure is out of hand or sometimes these servers are located in farthest countries when a problem arise, it is difficult to solve due to lack of integration and access of information. On the other hand, these systems need people who understand using digital platforms, but also having the back up on the health information, how data is securely preserved - I think when we are able to understand these digital dynamics then we can fully invest and operate all systems digitally…...."* (DMO 05)

## Discussion

The majority of primary perinatal data produced in the Kilimanjaro region are sourced from dispensaries which constitute over 80% of health facilities. The MTUHA books system operated by NMs in these facilities demands manual data entry into multiple records, as well as varying amounts of duplication, in the context of busy clinical environments. Additionally, data storage and retrieval are further challenges with books reported as sometimes becoming wet or torn, so, while in theory the high level of standardisation should provide robust primary data for health service planning and target setting, in practice data quality in monthly summary reports may be unreliable and/or unavailable. These findings concur with those of others [1,12,15,20], regarding weak data quality undermining efforts to improve outcomes in maternal and child health. Furthermore, it is of concern that sensitive personal health data, as categorized by the WHO, are not being managed with the stringent standards of safety and security they warrant.

Data loss and retrieval difficulties can also impact negatively on clinical care through unavailable historical information and inefficient use of staff time. Our NM participants described challenging work environments which made contemporaneous recording of data difficult, in line with a previous report [3]. However, they were found to be well engaged with the data system despite its challenges and spoke of utilising data recordings in their clinical role. This contrasts with the findings of Unkels *et al.* [3], in which nurse-midwives were found to be alienated from quantitative HMIS data systems, and not finding this type of data useful in their daily clinical work.

Among our participant contributions there is evidence of a culture of data-informed decision-making in clinical practice and at management level as advocated for LMICs by Unkels *et al.* [3], along with a desire for an improved system. The

positive perceptions and expectations across all staff groups regarding the introduction of a fully digitised health records system were striking. Participants expressed predominantly positive attitudes, with minimal reported reservations. They appear to be receptive users, which has been described by Karamagi *et al.* [11] as a key factor in the successful implementation of digital health interventions. We note however, that some of our participants stated that staff will quickly revert to paper registers if a digital system does not function in a manner which meets their and the health services' needs. Hesitancy in engaging with a digital data recording system, as reported elsewhere in sub-Saharan Africa [10,12], was not evident. This is likely influenced by some degree of knowledge and experience among participants of DHMIS through their work, along with the high penetration of mobile networks (>90%) in Tanzania [32], and high level (30%) of smartphone ownership amongst those who use mobile handsets [33]. Nonetheless, hesitancy was reported in a recent Tanzanian study [14] so the contrast in the findings between the studies is interesting. It may be explained by the difference in focus. Mwogosi and Kibusi's study [14] explored barriers experienced in EHR implementation, while this study sought reflections on the paper-based system, followed by gathering prospective perceptions and recommendations for a DHMIS. The differences may also reflect variation in professional roles, (un)familiarity with DMIS2 prior to the implementation of EHRs, or local conditions, e.g., leadership.

Several benefits of a full DHMIS were identified including, at clinical level, improved data recording and retrieval, along with an anticipated reduction in workload leading to time and cost efficiencies. In turn these were predicted to provide staff with more time to focus on delivering safer and improved services. At management level, it is considered to offer more rapid identification of emerging issues with the potential of speedier decisions and responses; it was described by one participant as a 'game changer' indicating expectations of a transformative impact. This reflects earlier work [6,7] concerning the feasibility and sustainability of such systems.

At the same time, staff across all four roles were not naïve about digital systems and were well informed about the potential sustainability challenges in low-resource settings including those related to electricity supply, network access and the need for technical support. When it came to making recommendations, perhaps unsurprisingly, the DMOs and DRCH-Cos who are involved in district level data management and are familiar with DHIS2, were most informed and vocal. They advocated for effective training across all relevant staff, expressing a desire for one integrated system to replace multiple existing systems, to achieve better health service efficiency and effectiveness; further, they highlighted the importance of investing in system sustainability at the stage of establishment and on an ongoing basis. These recommendations provide potential solutions to evident challenges and correspond closely with digital systems concerns reported elsewhere [13–16].

## Study limitations

This qualitative study is limited in that it covers a small percentage of over 400 healthcare sites in the Kilimanjaro region, and the number of Nurse Midwives, as the primary users of both paper and electronic systems, was quite limited. The use of one-to-one interviews provided rich data from experienced practitioners, though language translation from Kiswahili to English may have led to potential loss of nuance. While findings may not be generalised across Tanzania, the diversity of contexts within the sample enhances the transferability of insights. This study is the first to amplify the perspectives of four staff groups directly involved in the generation and use of data in the current Tanzanian HMIS and, as per Siggelkow [27], persuasion with case studies is possible when novelty is present. Further, it responds well to Kumar and Mustafa's call for in-depth research to offer guidance for HMIS improvements [2] and aligns with Mwogosi *et al.'s* (13) recommendation of participatory systems design.

The study also represents a single time period, November 2022 – March 2023. This did not allow for examination of shifts in attitudes as staff engaged with both the existing MTUHA system and experiences with new electronic systems. However, this limitation is mitigated by the long-term stability of the Tanzanian HMIS and its related MTUHA books. This may explain the striking level of consistency achieved across all four participant cohorts, with no major contradictions

noted. Future studies would benefit from examination of perceptions across more regions. Tanzania is a large country with a large healthcare staff cohort, and responses are likely to vary. A larger sample could be accommodated with a different study design such as a survey. However, as this study is formative and exploratory in nature, the semi-structured interview approach followed by thematic analysis, is considered appropriate.

## Conclusion

Quality health data is essential to improve the delivery and outcomes of healthcare services. This article reports on a qualitative case study with four groups of healthcare staff directly involved in collecting, recording, sharing and using perinatal data, within rural and urban settings in the largely rural Kilimanjaro region of Tanzania. The current hybrid paper and digital HMIS produces data which informs both clinical and policy decision making, however there are clear risks that some of the data are less than accurate, accessible, timely, or secure. The HMIS is therefore in need of reform, with digitalisation, as promoted by the WHO, being a feasible option.

Our study provides strong evidence of a high level of receptivity among four well-informed staff groups regarding full digitalisation of the HMIS. This, along with Tanzania's long-standing policy and practice of collecting standardised data on paper, means that two of the necessary pre-conditions for successful implementation of digital health interventions, as proposed by Karamagi *et al.* [11], are met. In the late 1990's Tanzania introduced a national HMIS to replace the disjointed systems used in different regions of the country and, just as 25 years ago, Tanzania has an opportunity to make a significant, evidence-informed advancement in the quality of maternal and child healthcare through implementing a standardised DHMIS. Challenges remain however, and electronic systems in Tanzania to date have been at best fragmented [21]. Nonetheless, Tanzania, through implementation of its national Digital Health Strategy [34], is also now making efforts to standardise and has made significant progress as measured against the WHO's Global Strategy on Digital Health [23]. If problems around knowledge sharing and capacity building [23] can be solved, Tanzania will be a leader in the African continent. Continued research in this area is needed, especially in the form of evaluation post implementation of a DHMIS. Dissemination of both a summary of this paper and presentations in Kiswahili will inform further reform work locally. Thanks to the securing of follow-on project funding, post implementation evaluations will be carried out in Tanzania as well as Uganda, Rwanda and Ethiopia over the next few years.

## Supporting information

**S1 File. Completed Questionnaire on Global Inclusivity.**
(DOCX)

## Acknowledgments

We wish to sincerely thank the healthcare staff who participated in our study for generously sharing their time, experiences, perceptions and recommendations.

## Author contributions

**Conceptualization:** Gaudensia A. Olomi, Modesta Mitao, Blandina T. Mmbaga, Jairy Khanga, Ali S. Khashan, Simon Woodworth.

**Data curation:** Mary Cronin, Lucy Munishi, Francis M. Pima, Simon Woodworth.

**Formal analysis:** Mary Cronin, Lucy Munishi, Jackline Somi, Francis M. Pima, Simon Woodworth.

**Funding acquisition:** Blandina T. Mmbaga, Ali S. Khashan.

**Investigation:** Lucy Munishi, Francis M. Pima.

**Methodology:** Mary Cronin, Ali S. Khashan, Francis M. Pima, Simon Woodworth.

**Project administration:** Mary Cronin, Lucy Munishi, Francis M. Pima, Simon Woodworth.

**Supervision:** Mary Cronin, Gaudensia A. Olomi, Modesta Mitao, Blandina T. Mmbaga, Ali S. Khashan, Francis M. Pima, Simon Woodworth.

**Validation:** Gaudensia A. Olomi, Modesta Mitao, Blandina T. Mmbaga.

**Writing – original draft:** Mary Cronin, Lucy Munishi, Francis M. Pima, Simon Woodworth.

**Writing – review & editing:** Mary Cronin, Lucy Munishi, Gaudensia A. Olomi, Modesta Mitao, Blandina T. Mmbaga, Jairy Khanga, Ali S. Khashan, Francis M. Pima, Simon Woodworth.

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
