## [Decision Letter · Decision Letter 0]

1 Jul 2025

Dear Dr. Cronin,

Thank you for submitting your manuscript to PLOS ONE. After careful consideration, we feel that it has merit but does not fully meet PLOS ONE’s publication criteria as it currently stands. Therefore, we invite you to submit a revised version of the manuscript that addresses the points raised during the review process.

Please submit your revised manuscript by  Aug 15 2025 11:59PM.  If you will need more time than this to complete your revisions, please reply to this message or contact the journal office at plosone@plos.org . A rebuttal letter that responds to each point raised by the academic editor and reviewer(s). You should upload this letter as a separate file labeled 'Response to Reviewers'.A marked-up copy of your manuscript that highlights changes made to the original version. You should upload this as a separate file labeled 'Revised Manuscript with Track Changes'.An unmarked version of your revised paper without tracked changes. You should upload this as a separate file labeled 'Manuscript'.

We look forward to receiving your revised manuscript.

Kind regards,

Hamufare Mugauri, Ph.D. Medicine and Health Sciences

Academic Editor

PLOS ONE

Journal Requirements:

Reviewers' comments:

Reviewer's Responses to Questions

**Comments to the Author**

1. Is the manuscript technically sound, and do the data support the conclusions?

Reviewer #1: Yes

Reviewer #2: Yes

2. Has the statistical analysis been performed appropriately and rigorously?

Reviewer #1: Yes

Reviewer #2: N/A

3. Have the authors made all data underlying the findings in their manuscript fully available?

Reviewer #1: Yes

Reviewer #2: Yes

4. Is the manuscript presented in an intelligible fashion and written in standard English?

Reviewer #1: Yes

Reviewer #2: No

Reviewer #1: Report

reviewer comment

Exploring healthcare staff experiences with a hybrid paper/digital health management information system and their

perspectives on digitalization as an alternative a Tanzanian qualitative case study on perinatal data

Suggestions for Improvement:

1. Improve literature review, historical background and add future scope

of study more effectively. To handle this issue, author can discuss some

well related and recent references.

2. Language quality needs improvement.

3. The references should have a clear and unified format .

4. I have reviewed the paper. Presentation, the related literature and tech nical soundness are good. It presents good method. After made some

additions to the paper, i recommend for publication.

5. Conclusion section should be enrich with some future remarks.

6. Some notations and equations are messed up or not clear. Please use a

proper equation editor format to make them clearly visible and under standable

7. all symbols should be italic in math form.

8. All the notations and abbreviations should be checked carefully.

9. In Keywords section add more keywords.

10. Are these new results sharp and more accurate compared with the others?

11. Please check and if needed add all the missing punctuation or missing

italics.

12. Some Theorem is not clear. Needs more discussion.

13. Please add (. and ,) in hole paper in suitable place.

14. I suggest adding some new relevant references to this topic.

(a) New forms of contra-continuity in ideal topology spaces. Missouri

Journal of Mathematical Sciences, 26(1):33–47, 2014.

2

(b) On almost e-I-continuous functions NT, Demonstratio Mathemat ica 54 (1),2021, 168-177

(c) Degree of (L, M)-fuzzy semi-precontinuous and (L, M)-fuzzy semi preirresolute functions,Demonstr. Math. 51 (1), 182197. Science

(d) Weak open sets on simple extension ideal topological space. Italian

Journal of Pure and Applied Mathematics, 2014, (33), pp. 333344

(e) Multi granulation on nano soft topological space. Advances in

Mathematics: Scientific Journal, 2020, 9(10), pp. 7711–7717.

(f) On topological groups via a-local functions, Applied General Topol ogy, 2014, 15(1), pp. 3342

(g) New types of multifunctions in ideal topological spaces via e-J open sets and δβ-J-open sets, Boletim da Sociedade Paranaense

de Matematica, 2016, 34(1), pp. 213223

Recommendation. I recommend accepting and publishing the paper after tak ing all the previous comments and observations and making some minor mod ifications mentioned above

Reviewer #2: Thanks for the opportunity to this research manuscript that addressed a contemporary issue on healthcare staff experiences with a hybrid paper/digital Health Management Information System (HMIS) in Tanzania. The authors have done wonderful job. Here is my review of the qualitative study;

Abstract

Strengths:

Clear description of objectives, methodology (case study, semi-structured interviews), and analytical approach (thematic analysis). Findings are logically summarised, highlighting key challenges with the hybrid system and positive perceptions of digitalisation.

Issues and Suggested Revisions:

The term "fully digital HMIS" appears repeatedly but could be more concise to improve readability.

The sentence:

“In part, because of these challenges, there is a strongly positive attitude among healthcare staff towards adopting a fully digital system…”

Could be simplified for flow:

“These challenges contribute to the strongly positive attitude among healthcare staff towards adopting a fully digital system…”

The recommendation section feels slightly compressed; splitting technical recommendations (training, integration, infrastructure) into clearer sentences would enhance readability.

Introduction

Strengths:

Well-structured with logical progression from global to national context and the use of recent WHO reports and region-specific literature strengthens the background.

Issues and Suggested Revisions:

Grammatical inconsistency noted here:

“while simultaneously, many factors are known to impede or promote investment…”

The phrase "while simultaneously" is redundant; recommend:

“At the same time, many factors are known to impede or promote investment…”

Some citations are dated or rely heavily on prior Tanzanian studies, though recent sources (2023-2024) are included, which is commendable.

The sentence:

"Hesitancy to transition fully to digital systems has been reported among healthcare staff mainly due to perceived risks..."

Could benefit from clarification:

“Hesitancy to transition to fully digital systems has been reported among healthcare staff, often attributed to concerns regarding sustainability and funding…”

Materials and Methods

Strengths:

Provides a thorough explanation of research design, sampling, and context. Reflexivity statement reflects good qualitative research practice, and the ethical considerations are clearly outlined.

Issues and Suggested Revisions:

Recruitment details imply purposive sampling but should explicitly state this.

Table 1 showing participant details is comprehensive, but one date seems inconsistent:

“NM 03 12/01/2022”

Should this be "12/01/2023" to align with other interview dates?

Under "Data Collection," this sentence could be clearer:

“Participants were provided with a small financial remuneration…”

Recommend specifying the amount or ethical justification for transparency.

Findings

Strengths:

The thematic structure is logical and comprehensive. Direct participant quotes enrich the narrative and provide credibility.

Issues and Suggested Revisions:

Some participant quotes are repetitive and could be trimmed for conciseness.

This sentence:

“Regardless of the level of facility, or its public, faith-run or private status, the types of data collected and recorded are highly standardised.”

Could be improved for flow:

“The types of data collected and recorded are highly standardised across facilities, irrespective of their public, faith-based, or private status.”

The transition between "Data Usage" and "Perceptions of a fully digitalised HMIS" could be made smoother with a bridging statement summarising challenges.

Discussion

Strengths:

Findings are linked back to existing literature, enhancing validity. Discusses both clinical and policy-level implications.

Issues and Suggested Revisions:

Repetition noted:

“These findings concur with those of Rumisha et al (1), Unkels et al(18), and Simba et al (13)…”

Reference numbers are not consistently spaced; apply consistent formatting.

This sentence:

“Several benefits of a fully digitised system were identified…”

Could be expanded briefly to include key examples for emphasis (e.g., improved data retrieval, workload reduction).

The claim:

“No negative attitudes were expressed, there was uniform support.”

Seems overly conclusive; recommend softening to:

“Participants expressed predominantly positive attitudes, with minimal reported reservations.”

Study Limitations

Strengths:

Acknowledges limitations of sample size and geographic focus. Recognises potential for generalisability limitations.

Issues and Suggested Revisions:

Contradiction:

“Constraints are mitigated by the diversity of contexts…”

Yet earlier states findings cannot be generalised. Suggest clarifying:

“While findings may not be generalised across Tanzania, the diversity of contexts within the sample enhances the transferability of insights.”

Recommend acknowledging language translation as a limitation due to potential loss of nuance from Kiswahili to English.

Conclusion

Strengths:

Aligns with WHO strategies and national health policy goals, and conveys optimism for digital transition.

Issues and Suggested Revisions:

This phrase:

“There is an opportunity again for Tanzania to take an evidence-based major leap forward…”

Is slightly informal for an academic paper; recommend:

“Tanzania has an opportunity to make a significant, evidence-informed advancement…”

Final sentences could emphasise the role of continued research and evaluation post-implementation.

References

Strengths:

Includes some recent and relevant sources (2023-2024).

Covers both global policy and Tanzanian-specific studies.

Issues and Suggested Revisions:

Some of the references can be upgraded using recent and relevant sources.

Inconsistent citation styles noted (e.g., missing author initials in some references).

Some hyperlinks (e.g., WHO document link) are incomplete or broken in the PDF format.

Reference [11]:

“medRxiv preprint” Given this is a preprint, recommend caution in interpretation and explicitly noting preprint status in the discussion if cited for evidence.

Summary of Major Recommendations:

Improve conciseness and readability in the Abstract by simplifying complex sentences.

Correct minor grammatical inconsistencies in the Introduction (e.g., redundant phrases).

Explicitly state purposive sampling under Methods and verify date inconsistencies in Table 1.

Reduce repetitive participant quotes in Findings and ensure smooth section transitions.

Rephrase overly conclusive statements in the Discussion for academic caution.

Clarify contradictions regarding generalisability in Study Limitations.

Adopt more formal, precise language in the Conclusion.

Standardise and verify reference formatting, ensuring all hyperlinks are functional.

**Do you want your identity to be public for this peer review?** For information about this choice, including consent withdrawal, please see our Privacy Policy

Reviewer #1: No

Reviewer #2: **Yes: ** Olasile Babatunde Adedoyin

---

## [Author Response · Author response to Decision Letter 1]

26 Sep 2025

Dear Dr. Mugauri and reviewers,

We have included our responses to the points by the reviewers in our Response to Reviewers file.

Regarding other points raised in the Decision Letter we have

(1) amended our manuscript to comply with PLOS ONE style guidelines

(2) submitted a completed Inclusivity in Global Health questionnaire

(3) included captions for our supporting information files in the manuscript.

I hope these satisfy your requirements.

Yours sincerely,

Mary Cronin (Corresponding Author)

---

## [Editor Report · Decision Letter 1]

15 Oct 2025

Exploring healthcare staff experiences with a hybrid paper/digital health management information system and their perspectives on digitalization as an alternative – a Tanzanian qualitative case study on perinatal data

PONE-D-25-08398R1

Dear Dr. Cronin,

We’re pleased to inform you that your manuscript has been judged scientifically suitable for publication and will be formally accepted for publication once it meets all outstanding technical requirements.

Kind regards,

Hamufare Dumisani Mugauri, Ph.D. Medicine and Health Sciences

Academic Editor

PLOS ONE
---

## [Editor Report · Acceptance letter]

PONE-D-25-08398R1

PLOS ONE

Dear Dr. Cronin,

I'm pleased to inform you that your manuscript has been deemed suitable for publication in PLOS ONE. Congratulations! Your manuscript is now being handed over to our production team.

Kind regards,

on behalf of

Dr Hamufare Dumisani Mugauri

Academic Editor

PLOS ONE